# Research on Repeated Quantum Games with Public Goods under Strong Reciprocity

Simo Sun [1,2,*], Yadong Shu [2], Jinxiu Pi [3] and Die Zhou [1]

1   School of Mathematics and Statistics, Guizhou University, Guiyang 550025, China; diezhou_gzu@126.com
2   School of Mathematics and Statistics, Guizhou University of Finance and Economics, Guiyang 550025, China; shuyadong@mail.gufe.edu.cn
3   School of Mathematics and Physics, Hunan University of Arts and Science, Changde 415000, China; pijinxiu@huas.edu.cn
*   Correspondence: sunsimo@mail.gufe.edu.cn

**Abstract:** We developed a repeated quantum game of public goods by using quantum entanglement and strong reciprocity mechanisms. Utilizing the framework of quantum game analysis, a comparative investigation incorporating both entangled and non-entangled states reveals that the player will choose a fully cooperative strategy when the expected cooperation strategy of the competitor exceeds a certain threshold. When the entanglement of states is not considered, the prisoner's dilemma still exists, and the cooperating party must bear the cost of defactoring the quantum strategy themselves; when considering the entanglement of states, the benefits of both parties in the game are closely related, forming a community of benefits. By signing a strong reciprocity contract, the degree of cooperation between the game parties can be considered using the strong reciprocity entanglement contract mechanism. The party striving to cooperate does not have to bear the risk of the other party's defector, and to some extent, it can solve the prisoner's dilemma problem. Finally, taking the public goods green planting industry project as an example, by jointly entrusting a third party to determine and sign a strong reciprocity entanglement contract, both parties can ensure a complete quantum strategy to maximize cooperation and achieve Pareto optimality, ultimately enabling the long-term and stable development of the public goods industry project.

**Keywords:** public goods; repeated game; strong reciprocity; quantum entanglement; Pareto

**MSC:** 91A12; 91A20; 91A81

## 1. Introduction

In 1954, Samuelson considered public goods as goods and services that can be shared by members of society in The Pure Theory of Public Expenditure.

A large number of public goods game experimental data summarized by Cardenas and Carpenter confirmed that human behavior deviated from the assumption of complete rationality [1]. Chen, Ye, and Wang pointed out that individuals were not all rational, and they had heterogeneous social preferences [2]. Dosi, G Marengo, L., and Fagiolo, G. proposed to examine the origins of institutional and evolutionary economics and to examine evolutionary economics theory from the perspectives of research guidelines and paradigms in order to further analyze the firm theory of evolutionary economics [3]. In terms of the supply and use of public goods or public resources, cooperation among individuals is ubiquitous, such as environmental protection, rural residents jointly raising funds to repair water canals, and urban residents jointly purchasing cleaning services. These phenomena are formed through participants' repeated or multiple-stage games. Cooperation strategy makes the collective benefit greater than the sum of individual benefits. Therefore, cooperation has synergy. In 1971, Friedman proved that each Nash equilibrium in which Pareto dominated the original game could be established in a perfect

equilibrium of repeated games [4]. Thereafter, Aumann and Shapley proposed to replace Nash equilibrium with subgame perfect equilibrium. Theoreticians have explained the reasons for the emergence and maintenance of cooperation from various perspectives. Robert believed that the repeated game of complete information was related to the evolution of the basic form of interactions between people, and proved that cooperation, altruism, revenge and threat in real life were the results of bounded rationality [5]. For the expectation of future interests in an infinitely repeated game, the most appropriate measure to promote cooperation is to use the theory of strong reciprocity [6]. Therefore, in the repeated game of public goods, we can promote and maintain cooperation by constructing a reasonable strong reciprocity mechanism. The SantaFe Institute is the main creator of the theory of strong reciprocity, and after more than a decade of development, the influence of the theory of strong reciprocity has become increasingly powerful. Gintis and Bowles is a trait of agents that operate even when there is no expected benefit from doing so [7,8]. Domestic scholars such as Wang, Ye, and Huang mainly introduce the research results of the Santafei Institute, without conducting in-depth research on strong reciprocity [9]. However, the papers by Gong Zhishui and Wei Qian review the new perspective of institutional evolution research on strong reciprocity in recent years [10,11]. Therefore, strong reciprocity is a strong form of cooperation.

Since the end of the 1990s, expanding domestic demand has been an important goal that the Chinese government is committed to pursuing, and the government has carried out a lot of work for it. However, the problem of insufficient domestic demand has not been fundamentally solved. In the context of the US financial crisis and the European debt crisis, reversing the traditional export-oriented economy and establishing an economic structure dominated by domestic demand becomes one of the central tasks of China's economic work. Studying the supply mechanism of public goods to promote a more adequate and effective supply of public goods can promote the expansion of domestic demand. Changing an export-oriented economy into a domestic demand economy is an important issue.

The COVID-19 that broke out at the end of 2019 caused heavy losses to social welfare. The externalities of the epidemic itself determine that public finance plays an important role in combating the epidemic. However, the issue of public finance construction during the epidemic reminds us that public goods cannot be completely provided by the market, and we should increase the diversification of public goods supply. To some extent, it is necessary to encourage enterprises to participate in the supply of public goods, so that they can pay attention to these social issues from the perspective of mechanism design, and it is also another business opportunity for enterprise development.

So far, research on games of public goods has mainly focused on theoretical analyses of market competitions based on Evolutionary Game with more than two populations [12] and the mechanism coexistence or supply chain mechanism [13,14].Yu,Yang et al. conducted a large amount of research on the equilibrium of population game [15–19], taking a multi-player repeated game in eBay online bidding as an example. Khakzad studied repeated games for eco-friendly flushing in reservoirs to study interactions between multiple self-interested parties [20]. Escobara and Llanes studied cooperation dynamics in repeated games of adverse selection to study cooperation dynamics in repeated games with Markovian private information. Many scholars have made a comparative analysis of the incentive effect of incentive contracts in different situations, and they have drawn corresponding conclusions and inspiration. Cao et al. believed that the government's incentive contract for manufacturers to recycle and remanufacture could better motivate manufacturers and also improve the government's revenue to a certain extent [21]. Tang, Dan, and Song pointed out that different discount rates would lead to different incentive effects of contracts [22]. So, the incentive effect of relationship contracts was gradually enhanced with the increase of discount rates.

In general, although some scholars have investigated and have discussed the incentive effects of different types of contracts, the binding force on game players is actually not strong. For example, free rider situations such as smart pig games still exist in public goods

games. Therefore, it is necessary to study the impact of strong reciprocity mechanisms on game players' strategies.

As the cross field of quantum mechanics and classical information theory, quantum information theory has played a significant role in promoting the development of quantum computers and has yielded a series of scientific research progressions. Quantum game theory is the product of the application of quantum information theory to the analysis framework of game theory and is also one of the new expansion fields of game theory. It was proposed in Meyer's paper on the quantum game of coin flipping [23]. Eisert and others further applied quantum game to the situation of the prisoner's dilemma [24]. Subsequent researchers in the field of physics and economics proposed relevant theorems, further enriching the quantum game theory [25]. Among them, Brandenburger compared and analyzed the difference between classical game and quantum game and pointed out that quantum strategy was not inferior to Nash equilibrium strategy [26]. Iqbal and Toor applied quantum games to the framework of evolutionary games and obtained an evolutionary stable strategy containing quantum strategies [27].

In recent years, scholars in information technology, computational mathematics, physics, electronic engineering and other disciplines have conducted very in-depth research on quantum games. Huang and Qiu investigated the quantized coward game and studied the influence of quantum decoherence on the Nash equilibrium solution of the quantum game [28]. Groisman proposed that quantum games could be regarded as classical extended games in some situations for quantized eagle pigeon games and prisoner's dilemma [29]. At the same time, domestic scholars in related fields have also made a series of studies. Zheng Junjun and others studied the exit dilemma caused by the different views of heterogeneous bidders and further solved the investment dilemma using quantum entanglement based on game theory. Yang and Zhang analyzed quantum repeated games with continuous-variable strategies [30]. Shi, Xu, and Chen analyzed the quantum Cournot duopoly game with an isoelastic demand function [31]. Wang and Yang analyzed the quantum equilibrium quantities and quantum equilibrium profits of nonlinear quantum Cournot duopoly games by using qualitative analysis [32] and quantum mixed duopoly games with a nonlinear demand function [33]. We believed in the advantages of quantum games that classical games could not achieve from the perspective of thinking form. Quantum games are a nonlinear, probabilistic, nondeterministic thinking mode. The research results showed that the greater the degree of entanglement of the game, the higher the overall maximum benefit. The above literature mainly focuses on the field of physics, but there are few articles on the economic background of applying quantum games to our production and life, and none on the applications of quantum game to the situation of repeated supply of public goods.

Quantum games without entanglement have the same outcome as classical games, and this type of problem may usually be solved using classical games of corresponding contracts.

In summary, from a new perspective of the game of repeated supply of public goods, this article first introduces a quantum strategy set and constructs a repeated quantum game model of public goods; secondly, using the paradigm of quantum game analysis, comparative research is conducted on whether to consider entangled states. Finally, an application analysis is conducted using the cooperation project of green planting industry in public goods greenhouses as an example.

## 2. Quantum Game Analysis of Repeated Supply of Public Goods under Strong Reciprocity

Public goods industry cooperation projects are not a non black or white binary strategic game of cooperation and non cooperation. Especially for public goods industry cooperation projects with long-term repetition, the degree of cooperative effort should be regarded as a continuous variable. This state, which exists between complete effort and complete non effort, is very similar to the concepts of superposition and the entanglement of states in

quantum mechanics. This article uses the analytical paradigm of quantum games, using a repeated game framework to study the evolution process of industrial cooperation in repeated quantum games of public goods with strong reciprocity mechanisms.

The public goods supply market consists of two populations: private enterprises and state-owned enterprises, with the supply strategy set of agents in each population being $\{C(\text{cooperate}), D(\text{defector})\}$. Then, the repeated game scenario of public goods supply is as follows.

Firstly, their investment is assumed to be the degree of effort $e_1$ and $e_2$, and the investment risks of game population are $E_c(e_1) = \frac{\gamma_1}{2}e_1^2$, $G_c(e_2) = \frac{\gamma_2}{2}e_2^2$, where $\gamma_1$ and $\gamma_2$ are the cost parameters of the two populations.

Secondly, since the mutual discount coefficient is strong and all populations are the same after the game starts in the second stage, we set the total market income $U$ of public goods to

$$U = be_1^\alpha e_2^{1-\alpha} + \varepsilon,$$

where $b$ denotes the coefficient of outputs; $\alpha$ represents the weight of the state-owned enterprise population's cooperation degree $1 - \alpha$; and $\varepsilon$ is a random perturbation on Cobb–Douglas.

In this article, the income distribution contract will temporarily consider the ordinary linear form, where $E(U) = \beta U$ and $\beta$ are the income distribution coefficients.

Further, let the degree of cooperation between private enterprise population and the state-owned enterprise population be $\theta_1 = 1 - e_1$ and $\theta_2 = 1 - e_2$, respectively, then $\theta_i = 0$ means full cooperation, and $\theta_i = 1$ means no cooperation. From the perspective of quantum games, the two polarized states of complete effort and complete no effort correspond to $\theta_i = 0$ and $\theta_i = 1$, respectively, and correspond to the two polarized quantum states of $|0\rangle$ and $|1\rangle$ in quantum theory. The corresponding return matrices are shown in Table 1.

**Table 1.** Payoff matrix of supply with public goods.

| Payoffs | The State-Owned Enterprises | |
|---|---|---|
| The private enterprises | $C(|0\rangle)$ | $D(|1\rangle)$ |
| $C(|0\rangle)$ | $(1-\beta)b - \frac{\gamma_1}{2}, \beta b - \frac{\gamma_2}{2}$ | $-\frac{\gamma_1}{2}, 0$ |
| $D(|1\rangle)$ | $0, -\frac{\gamma_2}{2}$ | $0, 0$ |

Thus, the payoff function of the private enterprises is

$$EE_U = (1-\beta)be_1^\alpha e_2^{1-\alpha} - E_c(e_1) = (1-\beta)b(1-\theta_1)^\alpha(1-\theta_2)^{1-\alpha} - \frac{\gamma_1}{2}(1-\theta_1)^2,$$

where $\beta$ represents the product factor of the return after the first game, and the payoff function of the state-owned enterprises is

$$EG_U = \beta be_1^\alpha e_2^{1-\alpha} - G_c(e_2) = \beta b(1-\theta_1)^\alpha(1-\theta_2)^{1-\alpha} - \frac{\gamma_2}{2}(1-\theta_2)^2.$$

Therefore, the benefits of pure strategies $(0, 0)$, $(0, 1)$, $(1, 0)$, and $(1, 1)$ for the private enterprises and the state-owned enterprises are $((1-\beta)b - \frac{\gamma_1}{2}, \beta b - \frac{\gamma_2}{2})$, $(-\frac{\gamma_1}{2}, 0)$, $(0, -\frac{\gamma_2}{2})$, and $(0, 0)$.

### 2.1. Repeated Game Based on Strong Reciprocity Public Goods

In the repeated game including private enterprises and the state-owned enterprises, these agents repeatedly play a game with public goods; the payoff matrix of such a game is put in Table 1.

We further make the following assumptions:

(i)  Game scenario strategy assumption: In the game process, both the private enterprises and the state-owned enterprises have only cooperation and betrayal strategies. After

the first game, a tit for tat update mechanism is adopted in the repeated game. In the sub game, the game population only has two strategy choices, that is, hypothesis $C$ strategy and $D$ strategy. We will not consider the escape strategy of the game population for the time being.

(ii) Game process parameter assumption: Consider the strong reciprocal punishment that affects the game strategy in the repeated game process as $\delta (0 < \delta < 1)$. So, when the player chooses the defector strategy, he will pay $\delta$ as a cost of defector. The payoffs on defector decreases with the increase in strong reciprocity $\delta$, which is reflected in the repeated quantum game payoffs $B_E$ and $C_G$ in 2.2 below.

(iii) Game result assumption: Consider the time value of returns in repeated game returns as $\rho (0 < \rho < 1)$. It is the discount factor and the probability of repeated games in the next stage. So, $1 - \rho$ is the probability of game ending.

According to the above assumptions, after the public goods supply game population conducts the first stage of the game, there are four kinds of returns from the second stage of the repeated game:

(1) In the second stage, we assume that the state-owned enterprises and private enterprises always choose to cooperate, except when opponents choose to defect, then the payoffs of private enterprises are $A_1 \sum_{i=0}^{\infty} \rho^i = \frac{A_1}{1-\rho}, (A_1 = (1-\beta)b - \frac{\gamma_1}{2})$; and the payoffs of the state-owned enterprises are $A_2 \sum_{i=0}^{\infty} \rho^i = \frac{A_2}{1-\rho}, (A_2 = \beta b - \frac{\gamma_2}{2})$.

(2) In the second stage, we assume that the state-owned enterprises with cooperation will always choose to defect in the future if the private enterprises respond by choosing to defect; then, the payoffs of the private enterprises are 0, and the payoffs of the state-owned enterprises are $\frac{C_2(1-\rho)-\delta\rho}{1-\rho}, (C_2 = \frac{\gamma_2}{2})$.

(3) This is similar to (2)—the private enterprises with cooperation will always choose to defect in the future if the state-owned enterprises respond by choosing to defect; then, the payoffs of the private enterprises are $\frac{B_1(1-\rho)-\delta\rho}{1-\rho}, (B_1 = \frac{\gamma_1}{2})$. and the payoffs of the state-owned enterprises are 0.

(4) In this repeated game, a tit for tat update strategy was adopted. Once a player chooses a defector strategy in the current stage, the other player will choose a defector (never cooperate) strategy in the following stages. That is to say, both sides of the game have chosen a defector strategy, and the payoffs on the game is 0.

### 2.2. Public Goods Repeated Quantum Game

In Sun et al. [34], the Pareto optimal state is achieved by designing a reasonable mechanism for strong reciprocity coefficient $\delta$ and discount factor $\rho$, but the achievement of the Pareto optimal state is roundabout. Fortunately, this problem is solved by some quantum schemes.

According to quantum game theory, the two polarization states $\theta_i = 0$ (complete cooperation) and $\theta_i = 1$ (complete betrayal) correspond to the polarized quantum states $|0\rangle$ and $|1\rangle$, respectively. Then, the corresponding payoff matrix is shown in Table 2.

**Table 2.** Payoff matrix of repeated supply of public goods based on quantum game.

| Payoffs | The State-Owned Enterprises | |
|---|---|---|
| The private enterprises | $C(|0\rangle)$ | $D(|1\rangle)$ |
| $C(|0\rangle)$ | $\frac{A_1}{1-\rho}, \frac{A_2}{1-\rho}$ | $-B_1 - \frac{\delta\rho}{1-\rho}, 0$ |
| $D(|1\rangle)$ | $0, -C_2 - \frac{\delta\rho}{1-\rho}$ | $0, 0$ |

The payoff matrix of Table 2 implies that the payoffs reach the Pareto optimal state only when the private enterprises and the state-owned enterprises choose the strategy of full effort. Moreover, if one cooperates completely and the other one betrays, then the fully

cooperative one will bear both the cost of the cooperation and no profit. This means the risk of the cooperation is aggravated by the increase in the public goods supply investment $(E_c, G_c)$. However, the Pareto optimal state is not the unique Nash Equilibrium.

According to the two extreme states $|0\rangle$ and $|1\rangle$, we set the initial quantum state of both parties as $|00\rangle$, where the first digit represents the private enterprises, the second digit represents the state-owned enterprises, and $|00\rangle = |0\rangle \otimes |1\rangle$. Then, let the entanglement matrix be

$$\widehat{J} = \exp(i\frac{w}{2}\sigma_x \otimes \sigma_x) = \cos\frac{w}{2} \cdot I + i\sin\frac{w}{2} \cdot H.$$

where

$$H = \begin{pmatrix} 0 & 0 & 0 & 1 \\ 0 & 0 & -1 & 0 \\ 0 & -1 & 0 & 0 \\ 1 & 0 & 0 & 0 \end{pmatrix},$$

$$\sigma_x = \begin{pmatrix} 0 & 1 \\ -1 & 0 \end{pmatrix},$$

$I$ denotes a $4 \times 4$ identity matrix, and $w$ represents the degree of entanglement. If $\omega = \frac{\pi}{2}$, namely, the degree of entanglement is the maximum, then the anti entanglement matrix is

$$\widehat{J}^+ = \cos\frac{w}{2} \cdot I - i\sin\frac{w}{2} \cdot H;$$

the strategy matrix of the private enterprises is

$$U_1(\theta_1, \varphi_1) = \begin{pmatrix} e^{i\varphi_1}\cos\frac{\theta_1}{2} & \sin\frac{\theta_1}{2} \\ -\sin\frac{\theta_1}{2} & e^{-i\varphi_1}\cos\frac{\theta_1}{2} \end{pmatrix};$$

and the strategy matrix of the state-owned enterprises is

$$U_2(\theta_2, \varphi_2) = \begin{pmatrix} e^{i\varphi_2}\cos\frac{\theta_2}{2} & \sin\frac{\theta_2}{2} \\ -\sin\frac{\theta_2}{2} & e^{-i\varphi_2}\cos\frac{\theta_2}{2} \end{pmatrix};$$

where $\theta_1, \theta_2 \in [0, \pi]$, $\varphi_1, \varphi_2 \in [0, \frac{\pi}{2}]$. The setup of a two-player quantum game is shown in Figure 1 [23].

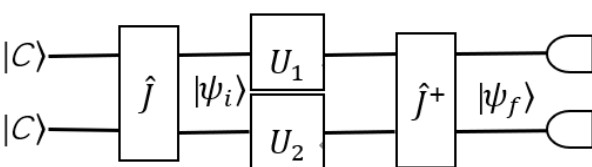

**Figure 1.** The setup of a two-player quantum game.

If $\theta_1 = 0$, $\varphi_1 = 0$, namely, both the private enterprises and the state-owned enterprise choose complete cooperation, then

$$U_1(0,0) = \begin{pmatrix} 1 & 0 \\ 0 & 1 \end{pmatrix}.$$

If $\theta_1 = \pi$, $\varphi_1 = 0$, namely, the private enterprises choose complete defector and the state-owned enterprise choose complete cooperation, then

$$U_1(\pi,0) = \begin{pmatrix} 0 & 1 \\ -1 & 0 \end{pmatrix}.$$

For the simplest entanglement-free situation, that is $\widehat{J} = I$, we have

$$|\psi_f\rangle = \widehat{J}^+(U_1 \bigotimes U_2)\widehat{J}|00\rangle = \begin{pmatrix} 0 & 0 & 1 & 0 \\ 0 & 0 & 0 & 1 \\ -1 & 0 & 0 & 0 \\ 0 & -1 & 0 & 0 \end{pmatrix} \cdot \begin{pmatrix} 1 \\ 0 \\ 0 \\ 0 \end{pmatrix} = -|10\rangle.$$

This degenerates to the situation of player 1's defector in the prisoner's dilemma game. When the entanglement of states is considered, namely, $\widehat{J} = \exp(i\frac{w}{2}\sigma_x \otimes \sigma_x)$, it follows that

$$
\begin{aligned}
|\psi_f\rangle &= \widehat{J}^+ \cdot [U_1(\theta_1, \varphi_1) \bigotimes U_2(\theta_2, \varphi_2)] \cdot \widehat{J}|00\rangle \\
&= [\cos(\varphi_1 + \varphi_2) - i \cdot \sin(\varphi_1 + \varphi_2)] \cos\frac{\theta_1}{2} \cos\frac{\theta_2}{2}|00\rangle \\
&+ [\cos\varphi_1 - i \cdot \sin\varphi_1 \cdot \cos w] \cos\frac{\theta_1}{2} \sin\frac{\theta_2}{2}|01\rangle \\
&+ [\sin w \cdot \sin\varphi_2] \sin\frac{\theta_1}{2} \cos\frac{\theta_2}{2}|01\rangle \\
&+ [\cos\varphi_2 - i \cdot \sin\varphi_2 \cdot \cos w]\sin\frac{\theta_1}{2} \cos\frac{\theta_2}{2}|10\rangle \\
&+ [\sin w \cdot \sin\varphi_1] \cos\frac{\theta_1}{2} \sin\frac{\theta_2}{2}|10\rangle \\
&+ [\sin w \cdot \sin(\varphi_1 + \varphi_2)] \cos\frac{\theta_1}{2} \cos\frac{\theta_2}{2}|11\rangle + \sin\frac{\theta_1}{2} \sin\frac{\theta_2}{2}|11\rangle.
\end{aligned}
$$

Thus, the probability of each quantum state is

$$
P(Q) = \begin{cases}
P_{00} &= [\cos^2(\varphi_1 + \varphi_2) + \sin^2(\varphi_1 + \varphi_2)\cos^2 w]\cos^2\frac{\theta_1}{2}\cos^2\frac{\theta_2}{2} \\
P_{01} &= [\cos^2\varphi_1 + \sin^2\varphi_1\cos^2 w]\cos^2\frac{\theta_1}{2}\sin^2\frac{\theta_2}{2} \\
&+ [\sin^2\varphi_2\sin^2 w]\sin^2\frac{\theta_1}{2}\cos^2\frac{\theta_2}{2} \\
P_{10} &= [\sin^2\varphi_1\sin^2 w]\cos^2\frac{\theta_1}{2}\sin^2\frac{\theta_2}{2} \\
&+ [\cos^2\varphi_2 + \sin^2\varphi_2\cos^2 w]\sin^2\frac{\theta_1}{2}\cos^2\frac{\theta_2}{2} \\
P_{11} &= [\sin^2(\varphi_1 + \varphi_2)\sin^2 w]\cos^2\frac{\theta_1}{2}\cos^2\frac{\theta_2}{2} + \sin^2\frac{\theta_1}{2}\sin^2\frac{\theta_2}{2}
\end{cases}.
$$

Obviously, since $P_{00} + P_{01} + P_{10} + P_{11} = 1$, the expected payoffs of the private enterprises are obtained as follows:

$$
\begin{aligned}
EE_U &= A_E P_{00} + B_E P_{01} + 0P_{10} + 0P_{11} \\
&= A_E[1 - \sin^2(\varphi_1 + \varphi_2)\sin^2 w]\cos^2\frac{\theta_1}{2}\cos^2\frac{\theta_2}{2} \\
&+ B_E[\cos^2\varphi_1 + \sin^2\varphi_1 \cdot \cos^2 w]\cos^2\frac{\theta_1}{2}\sin^2\frac{\theta_2}{2} \\
&+ B_E[\sin^2\varphi_2 \cdot \sin^2 w]\sin^2\frac{\theta_1}{2}\cos^2\frac{\theta_2}{2};
\end{aligned}
$$

and the expected payoffs of the state-owned enterprises are obtained as follows:

$$
\begin{aligned}
EG_U &= A_G P_{00} + 0P_{01} + C_G P_{10} + 0P_{11} \\
&= A_G[1 - \sin^2(\varphi_1 + \varphi_2)\sin^2 w]\cos^2\frac{\theta_1}{2}\cos^2\frac{\theta_2}{2} \\
&+ C_G[\sin^2\varphi_1 \cdot \sin^2 w]\cos^2\frac{\theta_1}{2}\sin^2\frac{\theta_2}{2} \\
&+ C_G[\cos^2\varphi_2 + \sin^2\varphi_2 \cdot \cos^2 w]\sin^2\frac{\theta_1}{2}\cos^2\frac{\theta_2}{2};
\end{aligned}
$$

where $A_E = \frac{A_1}{1-\rho}$, $B_G = \frac{A_2}{1-\rho}$, $B_E = -B_1 - \frac{\delta\rho}{1-\rho}$, $C_G = -C_2 - \frac{\delta\rho}{1-\rho}$.

## 3. Entanglement of Quantum States

Since the game process is uniformly affected by the entanglement $\omega$, we only need to discuss the entanglement with or without states.

### 3.1. The Entanglement without States

When the entanglement of states is not considered, i.e., $\omega = 0$, $\hat{J}^+ = \hat{J} = I$, we have

$$
\begin{aligned}
|\psi_f\rangle &= \hat{J}^+ \cdot [U_1(\theta_1, \varphi_1) \bigotimes U_2(\theta_2, \varphi_2)] \cdot \hat{J}|00\rangle \\
&= e^{i(\varphi_1+\varphi_2)} \cos\frac{\theta_1}{2} \cos\frac{\theta_2}{2}|00\rangle - e^{i\varphi_1} \cos\frac{\theta_1}{2} \sin\frac{\theta_2}{2}|01\rangle \\
&\quad - e^{i\varphi_2} \sin\frac{\theta_1}{2} \cos\frac{\theta_2}{2}|10\rangle + \sin\frac{\theta_1}{2} \sin\frac{\theta_2}{2}|11\rangle;
\end{aligned}
$$

the payoffs of the private enterprises are

$$
\begin{aligned}
EE_U &= A_E P_{00} + B_E P_{01} + 0P_{10} + 0P_{11} \\
&= A_E cos^2\frac{\theta_1}{2} \cos^2\frac{\theta_2}{2} + B_E \cos^2\frac{\theta_1}{2} \sin^2\frac{\theta_2}{2};
\end{aligned}
$$

and the payoffs of the state-owned enterprises are

$$
\begin{aligned}
EG_U &= A_G P_{00} + 0P_{01} + C_G P_{10} + 0P_{11} \\
&= A_G \cos^2\frac{\theta_1}{2} \cos^2\frac{\theta_2}{2} + C_G \sin^2\frac{\theta_1}{2} \cos^2\frac{\theta_2}{2}.
\end{aligned}
$$

**Theorem 1.** *For the entanglement without states,*

*(i) The private enterprises' payoffs $EE_U$ raise the increase in effort degree $e_1$ if and only if*
$A_E \cos^2\frac{\theta_2}{2} + B_E \sin^2\frac{\theta_2}{2} > 0$;
*(ii) the state-owned enterprises' payoffs $EG_U$ lower the increase in effort degree $e_2$ if and only if*
$A_G \cos^2\frac{\theta_1}{2} + C_G \sin^2\frac{\theta_1}{2} > 0$.

**Proof.** Since proof of the state-owned enterprises is similar to the private enterprises', we only verify the case of the private enterprises. By

$$
\begin{aligned}
EE_U &= A_E P_{00} + B_E P_{01} + 0P_{10} + 0P_{11} \\
&= A_E \cos^2\frac{\theta_1}{2} \cos^2\frac{\theta_2}{2} + B_E \cos^2\frac{\theta_1}{2} \sin^2\frac{\theta_2}{2} \\
&= \cos^2\frac{\theta_1}{2}(A_E \cos^2\frac{\theta_2}{2} + B_E \sin^2\frac{\theta_2}{2}),
\end{aligned}
$$

it follows that the private enterprises' payoffs $EE_U$ is positive if and only if $A_E \cos^2\frac{\theta_2}{2} + B_E \sin^2\frac{\theta_2}{2} > 0$. Hence, the private enterprises' payoffs $EE_U$ raise the increase in effort degree $e_1$.

So, Theorem 1 is proved.  □

In this game, the benefits of the players are equally affected by strong reciprocity and quantum entanglement. When conducting numerical simulation analysis, we only consider the impact of private enterprise profits. In order to better reflect the impact of strong reciprocity $\delta$ on the payoffs of the private enterprises, we first give the parameters $\beta = 0.4, b = 1.5, \gamma_1 = 0.2, \rho = 0.5$ and then take $\delta = 0, \delta = 0.4, \delta = 0.8$ for numerical simulation.

The numerical simulation diagrams are as follows:

Figure 2. Among them, (a) is a three-dimensional image of the enterprises' payoffs and effort level without quantum entanglement, (b) is $EE_U$ and $\theta_2$ corresponds to projection on $\theta_1$, (c) is $EE_U$, and $\theta_1$ corresponds to projection on $\theta_2$.

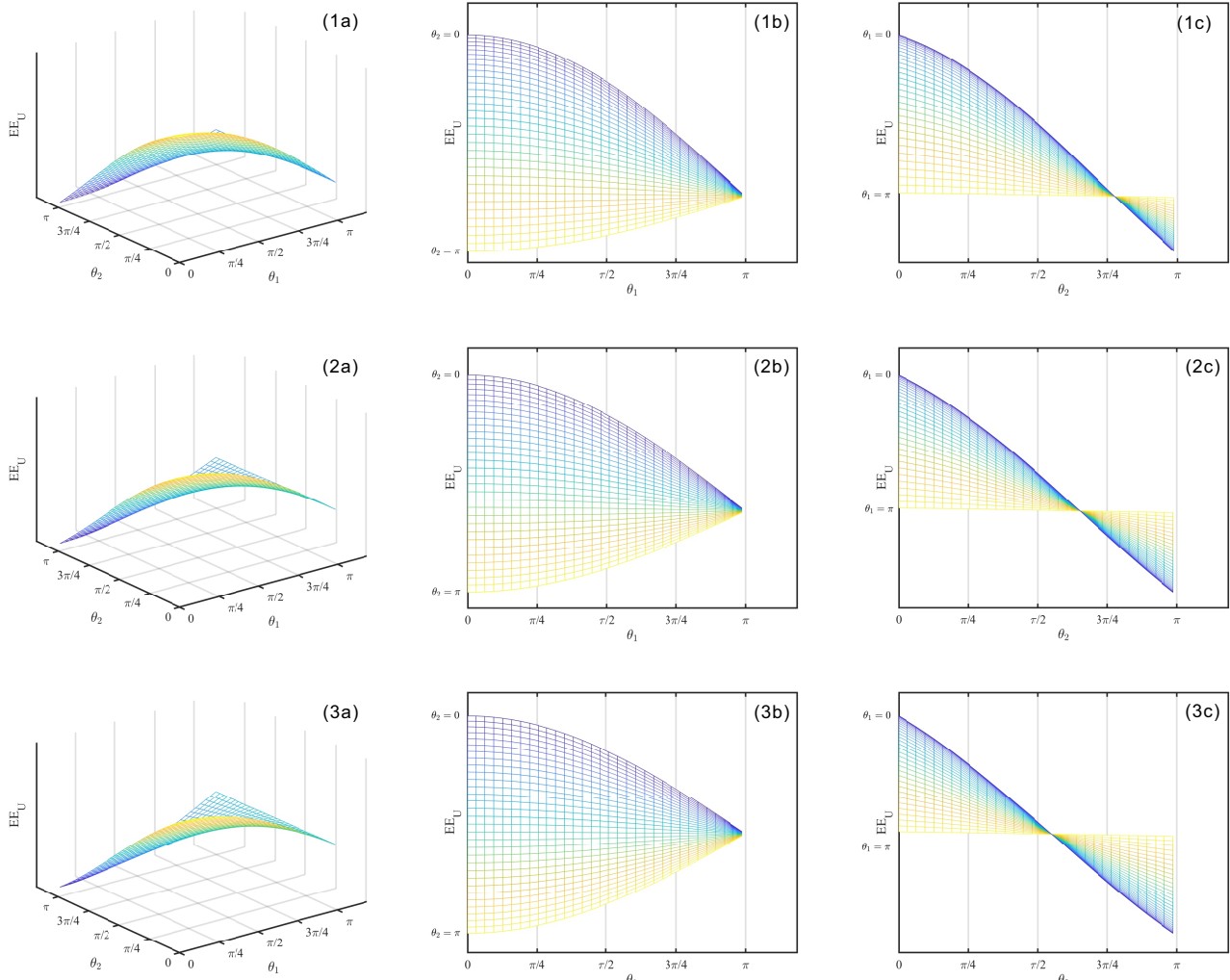

**Figure 2.** Three-dimensional image of the enterprises' payoffs $EE_U$ (**1a,2a,3a**). $EE_U$ and $\theta_2$ corresponds to projection on $\theta_1$ (**1b,2b,3b**); $EE_U$ and $\theta_1$ corresponds to projection on $\theta_2$ (**1c,2c,3c**).

From Figure 2(1a–3a), we find that the closer $\theta_2$ is to 0, the more obvious the decreasing trend of the private enterprises' payoffs $EE_U$ are with the increase in $\theta_1$. And as $\theta_2$ is close to $\pi$, it is difficult to judge whether the private enterprises' payoffs $EE_U$ will increase or decrease with $\theta_1$, because $A_E \cos^2 \frac{\theta_2}{2} + B_E \sin^2 \frac{\theta_2}{2} > 0$.

However, from the projections in Figure 2(1b–3b), we can see that when the strong reciprocity parameter takes different values of $\delta$, the payoffs $EE_U$ of private enterprises increase as $\theta_1$ increases from 0 to a certain threshold and $\theta_2$ increases, but after reaching the threshold, it decreases as $\theta_1$ increases from threshold to $\pi$ and $\theta_2$ increases.

At the same time, the threshold of $\theta_2$ is obtained by taking different values of $\delta$ for the positive and negative payoffs $EE_U$ of private enterprises. As mentioned earlier, when $\delta$ is 0, 0.4 and 0.8, respectively, the thresholds for $\theta_2$ are $arcsin\sqrt{\frac{8}{11}}$, $arcsin\sqrt{\frac{8}{13}}$, and $arcsin\sqrt{\frac{8}{15}}$. Therefore, it can be seen that the payoffs of the game party decrease with the increase in the value of the strong reciprocity parameter $\delta$, and this change is very clear in the numerical simulation of the threshold point $\theta_2$ in Figure 2(1c–3c).

Table 3 provides the payoff matrix of four special strategies for the private enterprises and the state-owned enterprises. From Table 3, we easily find the value of the parameter $\varphi_1$ and $\varphi_2$ do not affect the payoffs. Besides, the payoffs of the private enterprises will be reduced to $B_E$ if the private enterprises fully cooperates ($\theta_1 = 0$) and state-owned enterprises do not choose to cooperate ($\theta_2 = 0$). Namely, the cost of betrayal of the state-owned enterprises will be borne by the effortful one.

**Table 3.** Payoff Matrix under Four Strategies without Entanglement.

| Payoff | The State Owned Enterprises | | | |
|---|---|---|---|---|
| The private enterprises | $\theta_2 = 0, \varphi_2 = 0$ | $\theta_2 = 0, \varphi_2 = \frac{\pi}{2}$ | $\theta_2 = \pi, \varphi_2 = 0$ | $\theta_2 = \pi, \varphi_2 = \frac{\pi}{2}$ |
| $\theta_1 = 0, \varphi_1 = 0$ | $A_E, A_G$ | $A_E, A_G$ | $B_E, 0$ | $B_E, 0$ |
| $\theta_1 = 0, \varphi_1 = \frac{\pi}{2}$ | $A_E, A_G$ | $A_E, A_G$ | $B_E, 0$ | $B_E, 0$ |
| $\theta_1 = \pi, \varphi_1 = 0$ | $0, C_G$ | $0, C_G$ | $0, 0$ | $0, 0$ |
| $\theta_1 = \pi, \varphi_1 = \frac{\pi}{2}$ | $0, C_G$ | $0, C_G$ | $0, 0$ | $0, 0$ |

Next, let $q \equiv \cos^2 \frac{\theta_2}{2}$, then $1 - q = \sin^2 \frac{\theta_2}{2}$. Noticeably, the enterprises chooses the strategy of full effort, and the sufficient and necessary conditions can be rewritten by $A_E q + B_E(1 - q) > 0$. That is, the conclusion of the quantum game model is consistent with that of the deterministic evolutionary game model when the entanglement of states is not considered, Only when the expectation of the opponent's level of effort exceeds a certain threshold will the game player choose a complete effort strategy without considering entanglement, and the prisoner's dilemma remains unresolved. Even if the trustworthy party uses quantum strategy to defect, the cost still needs to be borne by the trustworthy party.

*3.2. The Entanglement of States*

Quantum game possesses extra state entanglements different from traditional game; this state entanglement has a positive effect on equilibria. For every state entanglement $\omega \in (0, \frac{\pi}{2})$, the probabilities of each quantum state are as follows:

$$
\begin{cases}
P_{00} = [\cos^2(\varphi_1 + \varphi_2) + \sin^2(\varphi_1 + \varphi_2)\cos^2 w]\cos^2 \frac{\theta_1}{2}\cos^2 \frac{\theta_2}{2} \\
P_{01} = [\cos^2 \varphi_1 + \sin^2 \varphi_1 \cos^2 w]\cos^2 \frac{\theta_1}{2}\sin^2 \frac{\theta_2}{2} + [\sin^2 \varphi_2 \sin^2 w]\sin^2 \frac{\theta_1}{2}\cos^2 \frac{\theta_2}{2} \\
P_{10} = [\sin^2 \varphi_1 \sin^2 w]\cos^2 \frac{\theta_1}{2}\sin^2 \frac{\theta_2}{2} + [\cos^2 \varphi_2 + \sin^2 \varphi_2 \cos^2 w]\sin^2 \frac{\theta_1}{2}\cos^2 \frac{\theta_2}{2} \\
P_{11} = [\sin^2(\varphi_1 + \varphi_2)\sin^2 w]\cos^2 \frac{\theta_1}{2}\cos^2 \frac{\theta_2}{2} + \sin^2 \frac{\theta_1}{2}\sin^2 \frac{\theta_2}{2}
\end{cases}
$$

Then, the expected payoffs of the private enterprises are

$$
\begin{aligned}
EE_U &= A_E P_{00} + B_E P_{01} + 0 P_{10} + 0 P_{11} \\
&= A_E[1 - \sin^2(\varphi_1 + \varphi_2)\sin^2 w]\cos^2 \frac{\theta_1}{2}\cos^2 \frac{\theta_2}{2} \\
&\quad + B_E[\cos^2 \varphi_1 + \sin^2 \varphi_1 \cdot \cos^2 w]\cos^2 \frac{\theta_1}{2}\sin^2 \frac{\theta_2}{2} \\
&\quad + B_E[\sin^2 \varphi_2 \cdot \sin^2 w]\sin^2 \frac{\theta_1}{2}\cos^2 \frac{\theta_2}{2};
\end{aligned}
$$

and the expected payoffs of the state-owned enterprises are

$$
\begin{aligned}
EG_U &= A_G P_{00} + 0 P_{01} + C_G P_{10} + 0 P_{11} \\
&= A_G[1 - \sin^2(\varphi_1 + \varphi_2)\sin^2 w]\cos^2 \frac{\theta_1}{2}\cos^2 \frac{\theta_2}{2} \\
&\quad + C_G[\sin^2 \varphi_1 \cdot \sin^2 w]\cos^2 \frac{\theta_1}{2}\sin^2 \frac{\theta_2}{2} \\
&\quad + C_G[\cos^2 \varphi_2 + \sin^2 \varphi_2 \cdot \cos^2 w]\sin^2 \frac{\theta_1}{2}\cos^2 \frac{\theta_2}{2}.
\end{aligned}
$$

We only consider the case of $\omega = \frac{\pi}{2}$ in the case that $0 < \omega < \frac{\pi}{2}$ is similar. Without special instructions, the entanglement of the considered state in this paper refers to $\omega = \frac{\pi}{2}$.

**Theorem 2.** *For the entanglement of states $\omega = \frac{\pi}{2}$,*

*(i) when the private enterprises adopt maximal quantum strategys ($\varphi_1 = \frac{\pi}{2}$) and $\sin^2 \varphi_2 \cos^2 \frac{\theta_2}{2} > 0$, the private enterprises' payoffs $EE_U$ raise the increase in effort degree $\theta_1$, $\delta$;*

*(ii) when the private enterprises adopt maximal non quantum strategy ($\varphi_2 = \frac{\pi}{2}$) and $\sin^2 \varphi_1 \cos^2 \frac{\theta_1}{2} > 0$, the state-owned enterprises' payoffs $EG_U$ raise the increase in effort degree $\theta_2$, $\delta$.*

**Proof.** We only consider the private enterprises due to the likeness of the state-owned enterprises.

Let $\varphi_1 = \frac{\pi}{2}$, then the private enterprises' payoffs $EE_U$ are

$$
\begin{aligned}
EE_U = &\left[A_E \cos^2(\varphi_1 + \varphi_2) \cos^2 \frac{\theta_2}{2}\right. \\
&\left. + B_E \cos^2 \varphi_1 \sin^2 \frac{\theta_2}{2}\right] \cdot \cos^2 \frac{\theta_1}{2} + B_E \sin^2 \varphi_2 \sin^2 \frac{\theta_1}{2} \cos^2 \frac{\theta_2}{2} \\
= &\left[A_E \cos^2 \frac{\theta_1}{2} + B_E \sin^2 \frac{\theta_1}{2}\right] \sin^2 \varphi_2 \cos^2 \frac{\theta_2}{2}.
\end{aligned}
$$

Obviously, since $A_E \cos^2 \frac{\theta_1}{2} + B_E \sin^2 \frac{\theta_1}{2}$ decreases with the increase in $\theta_1$, $\sin^2 \varphi_2 \cos^2 \frac{\theta_2}{2} > 0$ implies that $EE_U$ decreases with the increase in $\theta_1$.

So, Theorem 2 is proved. □

Noteworthy, the achievement of Theorem 2 has to join a third party to determine the strong reciprocal punishment and the time value of returns observable and quantifiable performance indicators and sign an "entanglement contract."

**Theorem 3.** *For the entanglement of states $\omega = \frac{\pi}{2}$,*

*(i) when the private enterprises adopts a non quantum strategy ($\varphi_1 = 0$) and $A_E \cos^2 \varphi_2 \cos^2 \frac{\theta_2}{2} - B_E \sin^2 \frac{\theta_2}{2} \geq 0$, the private enterprises' payoffs $EE_U$ raise the increase in effort degree $\theta_1$, $\delta$;*

*(ii) when the state-owned enterprises adopt the non quantum strategy ($\varphi_2 = 0$) and $A_G \sin^2 \varphi_1 \cos^2 \frac{\theta_1}{2} + C_G \sin^2 \frac{\theta_1}{2} > 0$, the the state-owned enterprises' payoffs $EE_U$ raise the increase in effort degree $\theta_2$, $\delta$.*

**Proof.** It is similar to the proof of Theorem 3, we only consider the private enterprises.

Let $\varphi_1 = 0$, then the private enterprises' payoffs $EE_U$ is

$$
\begin{aligned}
EE_U = &\left[A_E \cos^2(\varphi_1 + \varphi_2) \cos^2 \frac{\theta_2}{2}\right. \\
&\left. + B_E \cos^2 \varphi_1 \sin^2 \frac{\theta_2}{2}\right] \cdot \cos^2 \frac{\theta_1}{2} + B_E \sin^2 \varphi_2 \sin^2 \frac{\theta_1}{2} \cos^2 \frac{\theta_2}{2} \\
= &\left[A_E \cos^2 \varphi_2 \cos^2 \frac{\theta_2}{2} + B_E \sin^2 \frac{\theta_2}{2}\right] \cos^2 \frac{\theta_1}{2} + B_E \sin^2 \varphi_2 \sin^2 \frac{\theta_1}{2} \cos^2 \frac{\theta_2}{2}.
\end{aligned}
$$

Obviously, $A_E \cos^2 \varphi_2 \cos^2 \frac{\theta_2}{2} + B_E \sin^2 \frac{\theta_2}{2} > 0$ decreases with the increase of $\theta_1$, and $\sin^2 \varphi_2 \cos^2 \frac{\theta_2}{2} > 0$ decreases with the increase of $\theta_1$. Notably, for $A_E \cos^2 \varphi_2 \cos^2 \frac{\theta_2}{2} + B_E \sin^2 \frac{\theta_2}{2} \geq 0$ and $\sin^2 \varphi_2 \cos^2 \frac{\theta_2}{2} \geq 0$, if "=" is not taken at the same time, $EE_U$ decreases with the increase of $\theta_1$.

Hence, Theorem 3 is proved. □

**Theorem 4.** *For the entanglement of states $\omega = \frac{\pi}{2}$,*

*(i) when the private enterprises adopts a quantum strategy ($\varphi_1 \in (0, \frac{\pi}{2})$) and $A_E \cos^2(\varphi_1 + \varphi_2) \cos^2 \frac{\theta_2}{2} + B_E \cos^2 \varphi_1 \sin^2 \frac{\theta_2}{2} \geq 0$, the private enterprises' payoffs $EE_U$ raise the increase in effort degree $\theta_1$, $\delta$;*

*(ii) when the state-owned enterprises adopt a quantum strategy ($\varphi_2 \in (0, \frac{\pi}{2})$) and $A_G \cos^2(\varphi_1 + \varphi_2) \cos^2 \frac{\theta_1}{2} + C_G \cos^2 \varphi_2 \sin^2 \frac{\theta_1}{2} > 0$, then the state-owned enterprises' payoffs $EG_U$ raise the increase in effort degree $\theta_2$, $\delta$.*

**Proof.** Similarly, we consider the private enterprises.

From the private enterprises' payoffs

$$EE_U = [A_E \cos^2(\varphi_1 + \varphi_2) \cos^2 \frac{\theta_2}{2}$$
$$+ B_E \cos^2 \varphi_1 \sin^2 \frac{\theta_2}{2}] \cdot \cos^2 \frac{\theta_1}{2} + B_E \sin^2 \varphi_2 \sin^2 \frac{\theta_1}{2} \cos^2 \frac{\theta_2}{2},$$

It follows that the first term on the right of $EE_U$ decreases with the increase in $\theta_1$ as $A_E \cos^2(\varphi_1 + \varphi_2) \cos^2 \frac{\theta_2}{2} + B_E \cos^2 \varphi_1 \sin^2 \frac{\theta_2}{2} > 0$, and the second term on the right of $F$ decreases with the increase in $\theta_1$ as $\sin^2 \varphi_2 \cos^2 \frac{\theta_2}{2} > 0$. Notably, for $A_E \cos^2(\varphi_1 + \varphi_2) \cos^2 \frac{\theta_2}{2} + B_E \cos^2 \varphi_1 \sin^2 \frac{\theta_2}{2} \geq 0$ and $\sin^2 \varphi_2 \cos^2 \frac{\theta_2}{2} \geq 0$, if = is not taken at the same time, $EE_U$ decreases with the increase in $\theta_1$.

Thereby, Theorem 4 is proved. $\square$

Next, we verify that Theorems 2, 3 and 4 by simulation. The parameters of Theorem 1 remain unchanged, then we still take $\delta = 0, \delta = 0.4, \delta = 0.8$. To compare the changes in payoffs $EE_U$ of private enterprises using two quantum states, it shows in Figure 3 of $\varphi_1 = \frac{\pi}{2}$ and $\varphi_2 = 0$ and $\varphi_1 = \frac{\pi}{2}$ and $\varphi_2 = \frac{\pi}{2}$.

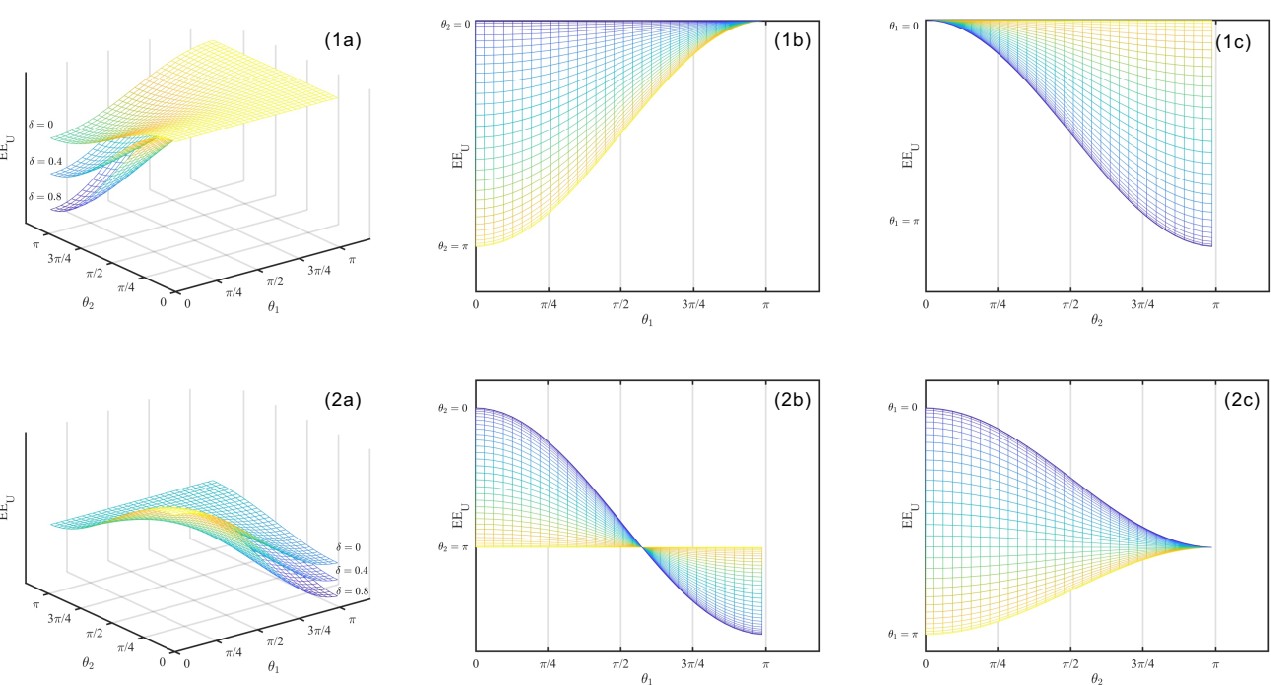

**Figure 3.** Three-dimensional image of $EE_U$ When taking $\varphi_1 = \frac{\pi}{2}$ and $\varphi_2 = 0$ (**1a**) or $\varphi_1 = \frac{\pi}{2}$ and $\varphi_2 = \frac{\pi}{2}$ (**2a**). $EE_U$ and $\theta_2$ corresponds to Projection on $\theta_2$ (**1b,2b**); $EE_U$ and $\theta_1$ corresponds to Projection on $\theta_2$ (**1c,2c**).

Firstly, from Figure 3(1a,2a), we can clearly see that when the strong reciprocity parameter $\delta$ takes different values, there is a significant change in the payoffs $EE_U$ of private enterprises. When $\theta_1$ and $\theta_2$ are closer to $\pi$, the payoffs $EE_U$ decrease with the increase in the strong reciprocity parameter $\delta$.

Secondly, from Figure 3(1b,2b,1c,2c), $\varphi_1 = \frac{\pi}{2}$, $\varphi_2 = 0$ and $\varphi_1 = \frac{\pi}{2}$, $\varphi_2 = \frac{\pi}{2}$, we find that if $\theta_1, \theta_2 \in (\frac{\pi}{4}, \frac{3\pi}{4})$, then the more obvious the trend is that the payoffs $EE_U$ of the private enterprise decreases with the increase in $\theta_2$ or $\theta_1$. Similarly, we find that if $\theta_2, \theta_1 \in (0, \frac{\pi}{4})$ or $\theta_2, \theta_1 \in (\frac{3\pi}{4}, \pi)$ it is difficult to judge the impact of the payoffs $EE_U$ of the private enterprises changing with $\theta_1$ or $\theta_2$. It implies that the private enterprise still tends to cooperate even if

the payoffs of the private enterprises decline due to the decline in the degree of cooperation between the state-owned enterprises.

Similar to Table 3, we also give the payoff matrix of private enterprises and the state-owned enterprises of four special strategies with the entanglement of states as shown in Table 4.

**Table 4.** Payoff matrix of repeated game for special strategic public goods.

| Payoffs | The State-Owned Enterprises | | | |
|---|---|---|---|---|
| The private enterprises | $\theta_2 = 0, \varphi_2 = 0$ | $\theta_2 = 0, \varphi_2 = \frac{\pi}{2}$ | $\theta_2 = \pi, \varphi_2 = 0$ | $\theta_2 = \pi, \varphi_2 = \frac{\pi}{2}$ |
| $\theta_1 = 0, \varphi_1 = 0$ | $A_E, A_G$ | 0,0 | $B_E, 0$ | $B_E, 0$ |
| $\theta_1 = 0, \varphi_1 = \frac{\pi}{2}$ | 0,0 | $A_E, A_G$ | $0, C_G$ | $0, C_G$ |
| $\theta_1 = \pi, \varphi_1 = 0$ | $0, C_G$ | $B_E, 0$ | 0,0 | 0,0 |
| $\theta_1 = \pi, \varphi_1 = \frac{\pi}{2}$ | $0, C_G$ | $B_E, 0$ | 0,0 | 0,0 |

According to Table 4, we easily find the full quantum strategy of maximum cooperation ($\theta_1 = 0$, $\varphi_1 = \frac{\pi}{2}$, $\theta_2 = 0$, $\varphi_2 = \frac{\pi}{2}$) achieves Pareto optimality and avoids the risk of defecting to the other one. Therefore, once private enterprises choose the maximum effort cooperation strategy, the defector of state-owned enterprises will result in their payoffs being damaged. If the state-owned enterprise industry does not adopt a quantum strategy, its profits will decrease to 0. If state-owned enterprises adopt a no-effort strategy, their profits will decrease to $C_G$, and they will bear the risk of defection. In the other three defector situations, if the private enterprise's payoffs only decrease to 0, then the private enterprise does not need to bear the risk of defection. Therefore, after considering the entanglement of states, it is only necessary to adopt a maximum-effort complete quantum strategy to achieve both self maximization and Pareto optimality, without the risk of the other party's defector.

To sum up, it is necessary to sign entanglement contracts in the game. The profits of both parties in the repeated game of public goods are closely related, forming a community of profits. By signing strong reciprocity entanglement contracts, the efforts of the game parties can be better measured using the strong reciprocity entanglement contract index, forming a good signal transmission for each other. The party who strives to cooperate does not need to bear the risk of the other party's defector; to some extent, it can solve the prisoner's dilemma problem of classical games.

## 4. Example Numerical Analysis

This section uses the analysis paradigm of quantum game to further analyze the game case of the repeated supply of public goods, vividly showing the theoretical analysis in Sections 3.1 and 3.2, and it further explains the way to promote the maximum cooperation between the state-owned enterprises and the private enterprises in the process of repeated supply of public goods.

Case: The private enterprise signed an agreement with the state-owned enterprise to jointly develop the distribution of the rural revitalization project "Greenhouse green vegetable planting industry in a certain place". The agreement stated that the state owned enterprises was responsible for land rent 35 needed for the "Greenhouse green vegetable planting industry", and the private enterprises were responsible for planting equipment, roads, irrigation and other infrastructure construction projects, as well as marketing. The total human cost was about 55. However, due to the existence of bilateral credit risk in the context of the greenhouse green vegetable planting industry, some hidden inputs will decline with the decline of the degree of cooperation between the two players. Suppose that the discount amount of the state-owned enterprises is between 30 and 60, and that of the private enterprises is between 40 and 80. Since the specific investment amount of both players cannot be observed by the other players, how can we promote long-term and repeated cooperation between both players?

Case analysis: This case is abstracted into a quantum game model of a repeated supply of public goods. Suppose that the strategy set of the private enterprises is $[40, 80]$, namely, $40 \leq e_1 \leq 80$, where $e_1$ is the cost of the enterprises. Similarly, let the strategy set of the state-owned enterprises be $[30, 60]$, then the cost of the state-owned enterprises $e_2 \in [30, 60]$. Let the final output meets $U = be_1e_2$ and the distribution coefficient is $\beta = 0.4$, then the relationship between strategy selection and effort level $\theta_1 = 2\pi - \frac{\pi}{40}e_1$, $\theta_2 = 2\pi - \frac{\pi}{30}e_2$.

This section mainly analyzes the operation and application of the public goods rural revitalization project "greenhouse green vegetable planting industry", mainly involving the concepts of quantum strategy and quantum entanglement. The difference between quantum strategy and classical strategy is that the imaginary unit $i$ is introduced, which is another dimension perpendicular to the real number axis in the coordinate system. In the application scenario of repeated supply of public goods, it is regarded as a measurable and quantifiable performance indicator in the rural revitalization project, such as total working hours, project construction costs, production costs, sales costs, etc. These observable indicators reflect the degree of cooperation of the players. Quantum strategies reflect both the degree of quantization $\varphi_i$ and the degree of cooperation $\theta_i$, which respectively represent various quantifiable performance indicators and the unquantifiable degree of cooperation of the agents. However, due to being incompletely positively correlated, the quantifiable performance indicators cannot fully reflect the degree of cooperation and relevant hidden investment in the project. So, it is necessary to strike an appropriate balance between the formulation of strong reciprocity policy and discount policy.

Quantum entanglement is to bundle the strategy of the agents together. It means the degree of correlation between the investment of agents is improved. A entanglement contract stating the project performance value with stronger reciprocal binding force should be signed before signing a contract for the repeated supply of public goods. Everyone sets a discount fund according to the stated performance value, and the amount of the discount fund is linked to the high or low performance target value. On the assessment date specified in the contract, the agents who fail to reach the performance target value in the assessment will be fined three times the difference in the discount fund. So, the investment of both agents will be tied. And, the previously stated performance target value can also convey some unobservable information. It greatly reduces the probability of credit risk behavior.

### 4.1. Regardless of Entanglement of States

In the first case, before the launch of the "Greenhouse green viable planting industry in a certain place" project, no relevant performance indicators were specified, that is, quantum strategy $\varphi_i$ and quantum entanglement $\omega$ were not considered. From the analysis of case 1 in Section 3.1, it is known that the expected payoffs of the private enterprises are

$$
\begin{aligned}
EE_U = {} & A_E \cos^2(\pi - \frac{\pi}{80}e_1) \cos^2(\pi - \frac{\pi}{60}e_2) \\
& + B_E \cos^2(\pi - \frac{\pi}{80}e_1) \sin^2(\pi - \frac{\pi}{60}e_2);
\end{aligned}
$$

and the expected payoffs of the state-owned enterprises are

$$
EG_U = A_G \cos^2(\pi - \frac{\pi}{80}e_1) \cos^2(\pi - \frac{\pi}{60}e_2) + C_G \sin^2(\pi - \frac{\pi}{80}e_1) \cos^2(\pi - \frac{\pi}{60}e_2).
$$

Let us first discuss the payoffs of private enterprises. According to the exported payoff function in Part 2, if the values of parameters $\beta = 0.4, b = 1.5, \gamma_1 = 0.2$ remain unchanged, then

$$EE_U = \frac{0.9e_1e_2 - 0.05e_1^2}{1 - \rho} \cos^2(\pi - \frac{\pi}{80}e_1)\cos^2(\pi - \frac{\pi}{60}e_2)$$
$$+ (e_1 - \frac{\delta\rho}{1 - \rho})\cos^2(\pi - \frac{\pi}{80}e_1)\sin^2(\pi - \frac{\pi}{60}e_2).$$

Under the model assumption in Section 3.1, we set $\rho = 0.5$, then

$$EE_U = (1.8e_1e_2 - 0.1e_1^2)\cos^2(\pi - \frac{\pi}{80}e_1)\cos^2(\pi - \frac{\pi}{60}e_2)$$
$$+ (e_1 - \delta)\cos^2(\pi - \frac{\pi}{80}e_1)\sin^2(\pi - \frac{\pi}{60}e_2)$$

Obviously, when $0 < \delta < 1$, that is to say, the game party has signed a strong reciprocity agreement, $e_1\cos^2(\pi - \frac{\pi}{80}e_1)$ is an increasing function of $e_1$.

When $\delta = 0$, the game becomes a one-time classic game. Then,

$$EE_U = (1.8e_1e_2 - 0.1e_1^2)\cos^2(\pi - \frac{\pi}{80}e_1)\cos^2(\pi - \frac{\pi}{60}e_2)$$
$$+ e_1\cos^2(\pi - \frac{\pi}{80}e_1)\sin^2(\pi - \frac{\pi}{60}e_2)$$

$e_1\cos^2(\pi - \frac{\pi}{80}e_1)$ is an increasing function of $e_1$.

Obviously, since $F(e_2) = (1.8e_1e_2 - 0.1e_1^2)\cos^2(\pi - \frac{\pi}{60}e_2) + e_1\sin^2(\pi - \frac{\pi}{60}e_2)$ is an increasing function of $e_2$, the increase or decrease of $EE_U$ is determined by the positive or negative of $F(e_2)$. Assume that the zero point of $F(e_2)$ is $e_2^*$, then,

(1) When $30 \leq e_2 \leq e_2^*$, $F(e_2)$ is negative, and $EE_U$ is a monotone decreasing function of $e_1$; the optimal choice of the private enterprises is $e_1^* = 40$, which is the minimum investment;

(2) When $e_2^* \leq e_2 \leq 60$, $F(e_2)$ is positive, and $EE_U$ is a monotone increasing function of $e_1$; the optimal choice of the private enterprises is $e_1^* = 80$, which is the maximum investment.

Similar to Theorem 1, when the strategy of state-owned enterprises reaches a certain threshold of $e_2^* \approx 42$, the optimal choice of private enterprises tends towards maximum cooperation.

### 4.2. Considering Entanglement of States

Now, we consider the entanglement of states. Before the launch of "Greenhouse green viable planting industry in a certain place" project, both parties entrusted to establish relevant measurable and quantifiable performance indicators and signed a binding entanglement contract. According to the analysis of Case 2 in Section 3.2, when the entanglement of states is being considered, the expected payoffs of the private enterprises are

$$EE_U = A_E[1 - \sin^2(\varphi_1 + \varphi_2)\sin^2 w]\cos^2\frac{\theta_1}{2}\cos^2\frac{\theta_2}{2}$$
$$+ B_E[\cos^2\varphi_1 + \sin^2\varphi_1 \cdot \cos^2 w]\cos^2\frac{\theta_1}{2}\sin^2\frac{\theta_2}{2}$$
$$+ B_E[\sin^2\varphi_2 \cdot \sin^2 w]\sin^2\frac{\theta_1}{2}\cos^2\frac{\theta_2}{2};$$

and the expected payoffs of the state-owned enterprises are

$$
\begin{aligned}
EG_U = {}& A_G[1 - \sin^2(\varphi_1 + \varphi_2)\sin^2 w]\cos^2\frac{\theta_1}{2}\cos^2\frac{\theta_2}{2} \\
& + C_G[\sin^2\varphi_1 \cdot \sin^2 w]\cos^2\frac{\theta_1}{2}\sin^2\frac{\theta_2}{2} \\
& + C_G[\cos^2\varphi_2 + \sin^2\varphi_2 \cdot \cos^2 w]\sin^2\frac{\theta_1}{2}\cos^2\frac{\theta_2}{2}.
\end{aligned}
$$

We keep the parameter values in 4.1 unchanged and we consider the maximum value of entanglement ($\omega = \dfrac{\pi}{2}$); we obtain the payoffs of the private enterprises, which are

$$
\begin{aligned}
EE_U = {}& (1.8e_1e_2 - 0.1e_1^2)\cos^2(\varphi_1 + \varphi_2)\cos^2\frac{\theta_1}{2}\cos^2\frac{\theta_2}{2} \\
& + (e_1 - \delta)\cos^2\varphi_1\cos^2\frac{\theta_1}{2}\sin^2\frac{\theta_2}{2} + (e_1 - \delta)\sin^2\varphi_2\sin^2\frac{\theta_1}{2}\cos^2\frac{\theta_2}{2};
\end{aligned}
$$

and the expected payoffs of the state-owned enterprises are

$$
\begin{aligned}
EG_U = {}& (1.2e_1e_2 - 0.1e_1^2)\cos^2(\varphi_1 + \varphi_2)\cos^2\frac{\theta_1}{2}\cos^2\frac{\theta_2}{2} \\
& + (e_1 - \delta)\sin^2\varphi_1\cos^2\frac{\theta_1}{2}\sin^2\frac{\theta_2}{2} + (e_1 - \delta)\cos^2\varphi_2\sin^2\frac{\theta_1}{2}\cos^2\frac{\theta_2}{2}.
\end{aligned}
$$

Here, it is assumed that the performance system of the private enterprises and the state-owned enterprises is determined by the total investment cost of the project implementation of the two populations. It is stipulated that the total investment of the private enterprises is 80, the total investment of the state-owned enterprises is 60, and the incentive fund for project performance is 50. Before signing the project contract, both parties signed a performance entanglement contract and agreed that once one party fails to reach the investment amount, he will be fined three times the difference in the incentive fund. Now, the quantization degree corresponds to the total investment, as shown in the following formula, $\varphi_1 = \frac{\pi}{2} \cdot \frac{E_c}{80}$ and $\varphi_2 = \frac{\pi}{2} \cdot \frac{G_c}{60}$.

Under the model assumption in Section 2.1, we set $\rho = 0.5$, $0 < \delta < 1$, and then the expected payoffs of the private enterprises are

$$
\begin{aligned}
EE_U = {}& (1.8e_1e_2 - 0.1e_1^2)\cos^2(\pi - \frac{\pi}{80}e_1)\cos^2(\pi - \frac{\pi}{60}e_2) \\
& + e_1\sin^2(\pi - \frac{\pi}{80}e_1)\cos^2(\pi - \frac{\pi}{60}e_2) \\
& - \delta(\pi - \frac{\pi}{80}e_1)\cos^2(\pi - \frac{\pi}{60}e_2);
\end{aligned}
$$

and the expected payoffs of the state-owned enterprises are

$$
\begin{aligned}
EG_U = {}& (1.2e_1e_2 - 0.1e_2^2)\cos^2(\pi - \frac{\pi}{80}e_1)\cos^2(\pi - \frac{\pi}{60}e_2) \\
& + e_2\cos^2(\pi - \frac{\pi}{80}e_1)\sin^2(\pi - \frac{\pi}{60}e_2) \\
& - \delta\cos^2(\pi - \frac{\pi}{80}e_1)\sin^2(\pi - \frac{\pi}{60}e_2).
\end{aligned}
$$

Therefore, we look at the payoffs of private enterprises: as long as $e_2 > \frac{5}{9}e_1$, and $cos^2(\pi - \frac{\pi}{60}e_2) \neq 0$, $\delta$ remains constant, the expected payoffs $EE_U$ of the private enterprises will increase with the increase in the degree of cooperation $e_1$. Similarly, we look at the payoffs of state-owned enterprises: as long as $e_1 > \frac{5}{6}e_2$, and $cos^2(\pi - \frac{\pi}{80}e_2) \neq 0$, $\delta$ remains constant, the expected payoffs $EG_U$ of the state-owned enterprises will increase with the increase in the degree of cooperation $e_2$. However, due to the existence of a strong

reciprocity coefficient $\delta$, it is easy to see that the expected payoffs of the game players will decrease with the increase in $\delta$. So, in the game of repeated supply of public goods, the strong reciprocity coefficient can promote the game population to repeat the supply of public goods. In this case, as long as the other party does not choose not to cooperate at all, the revenue of the game population will increase with the increase in the cooperation degree, instead of bearing the loss of the other party's defector. The signing of the entanglement contract can increase the constraints on both sides of the game and reduce the occurrence of free riding and other smart pig game phenomena.

**5. Conclusions**

We constructed a repeated quantum game model of strong-reciprocity public goods with quantum entanglement using the analytical paradigm of quantum games. Through comparative studies on entanglement without considering states and entanglement when considering states, a numerical analysis was conducted on greenhouse green planting industry cooperation projects under strong reciprocity. The following conclusions were drawn:

1. Because of quantifiable explicit discount indicators, the signing of strong reciprocity entanglement contracts can better improve the correlation of returns between the game parties in public goods industry cooperation projects and can better promote the cooperation efforts of the game parties. From the analysis in Section 3.2, after considering the entanglement of states, adopting a complete quantum strategy makes it easier to achieve an increase in returns with the increase in one's own efforts. The cost of defection by the other party is no longer borne by the striving party, which solves the prisoner's dilemma problem in classical games to some extent.

2. In the analysis in Section 3, due to the entanglement of states, the cooperative game parties in the public goods industry need to entrust a third party to determine the strong reciprocity entanglement contract index before the project implementation, which ensures that there is no motivation for the game parties to adopt non quantum strategies. Then, only the fully quantum strategy with the maximum degree of cooperation is optimal.

3. It is reasonable to analyze the repeated game of public goods from the perspective of quantum games. Due to the complexity of implementing cooperation projects in the public goods industry, the strategies of the game players are not a pure set of cooperative or non cooperative strategies. Because of many influencing factors, the degree of cooperation between the game players should be a continuous set of strategies, similar to the superposition of states in quantum games. The superposition of states and entanglement of states in quantum games can better reflect the cooperative degree of repeated games in public goods.

With the advancement of artificial intelligence, games in the future will transcend human–human interactions and encompass human–machine or machine–machine interactions. Furthermore, with the progress in quantum computing, quantum gaming is poised to become commonplace. By contemplating the evolution of cooperation under quantum entanglement and strong reciprocity mechanisms, we try to find more interesting game implementation mechanisms to solve the social cooperation dilemma.

**Author Contributions:** Conceptualization, S.S. and D.Z.; methodology, S.S.; software, Y.S.; validation, S.S., Y.S. and J.P.; formal analysis, J.P.; investigation, S.S.; resources, S.S.; data curation, Y.S.; writing—original draft preparation, S.S.; writing—review and editing, J.P.; visualization, D.Z.; supervision, D.Z.; project administration, S.S.; funding acquisition, S.S. All authors have read and agreed to the published version of the manuscript.

**Funding:** This research was funded by Universities Key Laboratory of System Modeling and Data Mining in Guizhou Province of funder grant number No.2023013.

**Data Availability Statement:** All data included in this study are available upon request by contact with the corresponding author.

**Conflicts of Interest:** The authors declare no conflict of interest.

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
