# Peer review of "Research on Repeated Quantum Games with Public Goods under Strong Reciprocity"

_axioms, doi:10.3390/axioms12111044_

Round 1

Reviewer 1 Report

Comments and Suggestions for Authors

All models are wrong but some are useful. It is nice that the idea of entanglement is used in economy modelling. Since the mechanisms of "entanglement" are much less "spooky" than in the case of particles in quantum mechanics. There are many ways by which the players may know about the action of the other. 

The authors apply the quantum formalism to classical prisoners dilemma game and achieve interesting results.

For a general reader, however, the conclusion part is very brief (and contains apparent typographic errors). The extension of the conclusion and, as well, a summary of comparison with results obtained by non-quantum approach, should be added. After that the article may be a very illustrative and useful. 

Reviewer 2 Report

Comments and Suggestions for Authors

The merit of this paper is that it explores new paths to model  systems in economics. Their approach is somewhat inspired to quantum physics within a line initiated by the Santa Fe group.  Therefore the merit is that in economics it is important it is important looking for new paths rather that taking  known tools of mathematics  with the hope that they can work also in economics.

The presentation should, however, be improved by  placing  the contents in a broader framework suitable to consider also the approach by mesoscopic models and the specific features of evolutionary/behavioral economics.

The author might refer to the book  G. Dosi, ``The Foundations of Evolutionary Economics’’, Oxford University Press and a review published in 2022 in the journal

``Physics of Life Reviews’’.

In my opinion, reference to the  Herbert A. Simon’s theory  of the artificial world , could contribute to the quality of the paper.  

Reviewer 3 Report

Comments and Suggestions for Authors

In this manuscript, authors investigate the role of entanglement and strong reciprocity in cooperation in quantum games. With the public goods in the focus, I think this manuscript has merits to be considered for a potential publication.

My concerns in this round that would require a revision are as follows.

An overall revision of the text is needed to fix some small typos, improve the readability, etc. Also, I think no need to write “Professor Groisman [28] of Cambridge University”, and something like only “Groisman [28]” would be both sufficient and be in accordance with the text. Citations to several works need to be fixed, such as not “Wang” but “Wang and Yang [29]”. Also, giving the citation at the end of the sentences could be better.

There are many recent achievements in quantum games in the literature, that the present text and the references need to be updated. Please check https://doi.org/10.3390/photonics9090617 and its references for up to date information.

Fig.1 can be considered the general process of quantum games –which I think is not very accurate- but anyways, it would require to cite Ref.[23] here as well.

Comments on the Quality of English Language

[copying from the report]

An overall revision of the text is needed to fix some small typos, improve the readability, etc. Also, I think no need to write “Professor Groisman [28] of Cambridge University”, and something like only “Groisman [28]” would be both sufficient and be in accordance with the text. Citations to several works need to be fixed, such as not “Wang” but “Wang and Yang [29]”. Also, giving the citation at the end of the sentences could be better.

Round 2

Reviewer 2 Report

Comments and Suggestions for Authors

Ok for the revision, just check spacing of a comma in the last lines of page 2

Reviewer 3 Report

Comments and Suggestions for Authors

The authors have successfully revised their manuscript except for one item about presenting and discussing the recent literature sufficiently.

I think this can be improved in a minor revision.